# Uridine Derivatives: Synthesis, Biological Evaluation, and In Silico Studies as Antimicrobial and Anticancer Agents

**DOI:** 10.3390/medicina59061107

**Published:** 2023-06-07

**Authors:** Nasrin S. Munia, Mohammed M. Alanazi, Youness El Bakri, Ashwag S. Alanazi, Yousef E. Mukhrish, Imtiaj Hasan, Sarkar M. A. Kawsar

**Affiliations:** 1Laboratory of Carbohydrate and Nucleoside Chemistry (LCNC), Department of Chemistry, Faculty of Science, University of Chittagong, Chittagong 4331, Bangladesh; nasrinsmunia@gmail.com; 2Department of Pharmaceutical Chemistry, College of Pharmacy, King Saud University, P.O. Box 2457, Riyadh 11451, Saudi Arabia; mmalanazi@ksu.edu.sa; 3Department of Theoretical and Applied Chemistry, South Ural State University, Lenin prospect 76, Chelyabinsk 454080, Russia; yns.elbakri@gmail.com; 4Department of Pharmaceutical Sciences, College of Pharmacy, Princess Nourah bint Abdulrahman University, P.O. Box 84428, Riyadh 11671, Saudi Arabia; asalanzi@pnu.edu.sa; 5Department of Chemistry, Faculty of Science, Jazan University, P.O. Box 2097, Jazan 45142, Saudi Arabia; yemukhrish@jazanu.edu.sa; 6Department of Biochemistry and Molecular Biology, Faculty of Science, University of Rajshahi, Rajshahi 6205, Bangladesh; hasanimtiaj@yahoo.co.uk

**Keywords:** uridine derivatives, antimicrobial agents, antiproliferative, molecular docking, molecular dynamics, cytotoxicity, pharmacokinetics

## Abstract

Nucleoside analogs are frequently used in the control of viral infections and neoplastic diseases. However, relatively few studies have shown that nucleoside analogs have antibacterial and antifungal activities. In this study, a fused pyrimidine molecule, uridine, was modified with various aliphatic chains and aromatic groups to produce new derivatives as antimicrobial agents. All newly synthesized uridine derivatives were analyzed by spectral (NMR, FTIR, mass spectrometry), elemental, and physicochemical analyses. Prediction of activity spectra for substances (PASS) and in vitro biological evaluation against bacteria and fungi indicated promising antimicrobial capability of these uridine derivatives. The tested compounds were more effective against fungal phytopathogens than bacterial strains, as determined by their in vitro antimicrobial activity. Cytotoxicity testing indicated that the compounds were less toxic. In addition, antiproliferative activity against Ehrlich ascites carcinoma (EAC) cells was investigated, and compound **6** (2′,3′-di-*O*-cinnamoyl-5′-*O*-palmitoyluridine) demonstrated promising anticancer activity. Their molecular docking against *Escherichia coli* (1RXF) and *Salmonella typhi* (3000) revealed notable binding affinities and nonbonding interactions in support of this finding. Stable conformation and binding patterns/energy were found in a stimulating 400 ns molecular dynamics (MD) simulation. Structure–activity relationship (SAR) investigation indicated that acyl chains, CH_3_(CH_2_)_10_CO-, (C_6_H_5_)_3_C-, and C_2_H_5_C_6_H_4_CO-, combined with deoxyribose, were most effective against the tested bacterial and fungal pathogens. Pharmacokinetic predictions were examined to determine their ADMET characteristics, and the results in silico were intriguing. Finally, the synthesized uridine derivatives demonstrated increased medicinal activity and high potential for future antimicrobial/anticancer agent(s).

## 1. Introduction

Pyrimidine nucleoside, that is, uridine (**1**), is an essential natural nucleoside that is a key component of ribonucleic acid (RNA). This compound serves as a foundational precursor to numerous biological molecules and plays a crucial role in the pyrimidine metabolic pathway within the central nervous system. Nucleoside derivatives are drugs synthesized from the modification of ribose or 2′-deoxyribose nucleosides [1,2,3,4]. Nucleoside derivatives are crucial drugs in the clinic for the treatment of cancer and viral infections [5,6,7]. Intracellular nucleoside analogs are activated by kinases to the active triphosphate metabolite, which is then inserted into deoxyribonucleic acid (DNA) and ribonucleic acid (RNA). This insertion leads to inhibition of viral replication and reduced cancer cell proliferation. Some studies have explored the efficacy of nucleoside analogs as antimicrobial agents [8,9]. Nucleoside analogs typically exhibit antiviral efficacy by reducing viral replication via cellular division, DNA/RNA synthesis impairment, or inhibition of cellular or viral enzyme activity [10]. However, numerous clinically approved pyrimidine and purine derivatives, including halogenated, azolated, and acylated derivatives, have antimicrobial activity. 2′-Deoxynucleosides such as trifluridine have been used to treat herpes virus infections [11,12], telbivudine (hepatitis B inhibitor) is also used to treat numerous viral disorders, and zidovudine is one of the most effective 2,3-dideoxynucleosides against HIV [13] Figure 1). Two effective nucleoside inhibitors of bacteria are 9-,β-D-arabinofuranosyladenine and 2′,3′-dideoxyadenosine. Hubert-Habart and Cohen reported the lethality of the former to a purine requiring a strain of Escherichia coli B. In this organism, 9-4-D-arabinofuranosyladenine markedly inhibited DNA synthesis and had virtually no effect on RNA synthesis. In addition, 2′,3′-dideoxyadenosine was shown to be lethal to selected strains of *Escherichia coli* by irreversibly inhibiting DNA synthesis in susceptible microorganisms.

Structurally, a nucleoside derivative is a nucleobase joined with a sugar ring with modifications different from those of natural nucleosides. Nucleobase modifications include acylation, halogenation, and the addition of azido groups, and sugar modifications include ring opening, halogenation, acylation, and hydroxylation or dehydroxylation [14,15,16]. In the sugar component, the modifications are mainly ribofuranose or deoxyribofuranose of nucleosides, including variation in the sugar component, alteration of the oxygen with another atom, insertion of heteroatoms in the sugar molecule, ring size alterations, and replacement with acyclic components [17]. Due to their chemical and physical properties, these variations result in significant alterations in biological behavior and toxicity dimensions [18]. Uridine, along with nucleobases, has been identified by researchers as being a pharmacologically varied category that includes cytotoxic compounds, antibacterial and antifungal drugs, and immune-suppressing molecules [19,20]. This discovery was made possible by the fact that nucleobases also contain uridine. Furthermore, studies on the regioselective acylation of nucleoside or carbohydrate derivatives and their subsequent antimicrobial screening [21,22] have demonstrated that the incorporation of two or more highly electron-dense heteroaromatic nuclei and aliphatic chains can significantly amplify the antimicrobial properties of the original nucleus [23].

In this study, novel uridine derivatives are designed and synthesized as antimicrobial agents based on the abovementioned features. Here, we report the in vitro antimicrobial tests of six uridine-based derivatives comprising aliphatic chains and aromatic rings against seven human and plant pathogens, molecular docking along with PASS prediction, and molecular dynamics study for the first time. In addition, density functional theory (DFT) was utilized to optimize the uridine derivatives that were synthesized, as well as to predict pharmacokinetic and drug-likeness characteristics.

## 2. Results

### 2.1. Chemistry and Characterization

This study aimed to design and synthesize novel uridine derivatives as therapeutic agents. The synthesis of compounds starts with a direct acylation method with regioselective palmitoylation of uridine **1** by palmitoyl chloride. Using an assortment of acylating agents, several derivatives of the palmitoylation product were synthesized. In this investigation, a number of compounds, including starting compound **1**, palmitoate **2**, and its derivatives **3**–**7**, were utilized as test compounds to evaluate their antibacterial and antifungal properties against various human pathogenic bacteria and plant pathogenic fungi.

Thus, we attempted the acylation of uridine **1** with unimolecular palmitoyl chloride in dry pyridine at −5 °C. Palmitoyl derivative **2** was used in the next stage after conventional work-up, solvent removal, and silica gel column chromatographic purification provided a 45% yield as crystalline solid (m.p. = 64–65 °C). Compound **2** was pure enough for subsequent reactions. FTIR, ^1^H-NMR, and elemental data determined the structure of palmitoyl derivative **2**. The sterically less inhibited primary hydroxyl group of the ribose moiety of uridine **1** is more reactive, forming 5′-oxo-palmitoyluridine **2** [24,25]. This compound’s structure was 5′-oxo-palmitoyluridine **2** after FTIR, ^1^H-NMR, and elemental data analysis (Section 3.2). Subsequently, palmitoyl product **2** was subjected to derivatization using two fatty acid chlorides. Compound **2** was treated with lauroyl chloride and afterwards underwent standard work-up procedures, resulting in the production of compound **3**. The yield of compound **3** was 56.98%, and it formed needle-shaped crystals with a melting point of 58–60 °C. The compound denoted as 2′,3′-di-oxo-lauroyl-5′-oxo -palmitoyluridine **3** was determined to possess two lauroyl groups attached at positions 2′ and 3′ through a thorough investigation of its FTIR, ^1^H-NMR, and elemental data. The 2′,3′-di-oxo-myristoyl-5′-oxo-palmitoyluridine **4** derivative of palmitoylation product **2** was readily synthesized utilizing the corresponding fatty acid chloride. FTIR and ^1^H-NMR spectra were thoroughly analyzed for characterization purposes. Figure 1 describes an extensive approach to the synthesis of MBG derivatives.

FTIR, ^1^H-NMR, and elemental data determined the structure of trityl derivative **5**. This compound’s FTIR spectra exhibited carbonyl stretching at 1703 cm^−1^. Two characteristic peaks in its ^1^H-NMR spectra were attributed to the compound’s two three-phenyl protons of trityl groups. The two trityl groups at positions 2′ and 3′ were attached by deshielding H-2′ and H-3′ protons to 5.41 and 5.38 (as m) from their precursor values (~4.00 ppm) [26]. FTIR, ^1^H-NMR, and elemental data determined this compound’s structure as 5′-oxo-palmitoyl-2′,3′-di-oxo-trityluridine, **5**. Encouraged by the results thus far, we treated compound **2** with cinnamoyl chloride in dry pyridine and silica gel chromatographic purification to produce the crystalline solid (m.p. = 59–61 °C) substitution product **6** in 64.96% yield. The FTIR spectra of compound **6** showed an absorption band at 1701 cm^−1^ for -C=O stretching. In the ^1^H-NMR spectra, two one-proton doublets at δ 7.57, 7.54 (2 × 1H, 2 × d, J = 16.0 Hz, 2 × PhC*H*=CHCO-) and two one-proton doublets at δ 6.0, 5.97 (2 × 1H, 2 × d, J = 16.0 Hz, 2 × PhCH=C*H*CO-) due to the presence of two cinnamoyl groups in the compound. Due to the two aromatic ring protons, there are also four- and six-proton multiplets at 8.10 and 7.28, respectively. The resonances of H-2′ (5.36) and H-3′ (5.34) in precursor compound **2** were downfield from their typical values, indicating the existence of cinnamoyl groups at positions 2′ and 3′. FTIR, 1H-NMR, and elemental data determined this compound’s structure as 2′,3′-di-oxo-cinnamoyl-5′ -oxo-palmitoyluridine, **6**.

The preparation of 4-*t*-butylbenzoyl derivative **7** revealed the structures of compound **2**. The reaction of component **2** with 4-*t*-butylbenzoyl chloride in dry pyridine produced derivative **7**. This chemical displayed carbonyl stretching absorption at 1718 cm^−1^ in the FTIR spectra. In its ^1^H-NMR spectra, two four-proton multiplets at 7.85 (2 × Ar-2H) and 7.51 (2 × Ar-2H) and two singlets at 1.38 and 1.36 {18H, 2 × s, 2 × (CH_3_)_3_C-} indicated two 4-*t*-butylbenzoyl groups. The 4-*t*-butylbenzoyl groups at positions 2′ and 3′ were shown by the downfield shift of H-2′ to 5.73 and H-3′ to 5.54 from their precursor diol values, **2**. The structure of this compound, 2′,3′-di-oxo- (4-*t*-butylbenzoyl)-5′-oxo-palmitoyluridine (**7**), was confirmed by FTIR and ^1^H-NMR spectra. Thus, many acylating agents have been used to prepare acylated uridine derivatives. These acyl chlorides were chosen for their physiologically prone atoms/groups to identify biologically active uridine derivatives.

### 2.2. Two-Dimensional NMR Analysis

The reduced field and finely allocated protonic signal from the Ar-NH displayed the first point in the COSY data of derivative **2**. Thus, Ar-NH’s peak at the diagonal’s middle left had a cross-peak marked Ar-NH, H-5′a, and H-5′b that linked to H-5′s signal. Thus, the Ar-NH proton at approximately δ 9.00 was coupled to the proton whose signal appeared at approximately δ (6.01 to 5.46) (i.e., H-5′a and H-5′b protons). Similarly, the peak of H-2′ was coupled to the signal from 2H, CH_3_(CH_2_)_13_CH_2_CO- by a cross-peak to explore the interaction between H-2′ and 2H, CH_3_(CH_2_)_13_CH_2_CO-. The lower field shift of H-1′, H-4′, H-3′, H-5′a, and H-5′b compared to initial derivative **2** ensured palmitoyl group insertions at C-5′ (Table 1). The structure of 5′-oxo-(palmitoyl)uridine was confirmed by examining its COSY, HSQC, and HMBC spectral experiments (Figure 2) and ^13^C-NMR spectrum (**2**).

### 2.3. Antibacterial Activity Analyses

Table 2 shows the antibacterial evaluation of the synthesized uridine derivatives (Figure 1) against five bacterial pathogens (Appendix A). According to the test results, compounds **4** and **5** showed a significant inhibition zone against both *Bacillus subtilis* (17 ± 0.20 mm and 13 ± 0.30) and *Bacillus cereus* (15 ± 0.50 mm and 16 ± 0.20), respectively (Appendix A). Compounds **1**, **2**, **3**, **6**, and **7** were inactive against Gram-positive organisms. Furthermore, compound **4** showed promising results against *S. typhi* (18 ± 0.75 mm). In the case of *E. coli*, compound 4 also showed good inhibition, but a moderate inhibition zone was obtained for compound **5** (10 ± 0.75 mm). Finally, no inhibition was observed for compounds **1**, **2**, and **6** among all bacterial types. Compounds **3**, 4, and **7** exhibited reasonable inhibition against the Gram-negative bacteria *P. aeruginosa* and *S. typhi* (Appendix A), which is compatible with our previous investigations [27].

### 2.4. MIC and MBC Screening

Based on the antibacterial results, compounds **4** and **5** showed considerable efficacy against all microorganisms. In support of this, compounds **4** and **5** were screened to further explore the MIC and MBC. The MIC values showed that compounds **4** and **5** had a maximum MIC value of 4.00 mg/L against *E. coli* and *B. subtilis* and a minimum value of 0.13 mg/L against *B. cereus*. The graphical representation of the MIC test (as presented in Appendix A) appears in Figure 3.

After determining the MIC for compounds **4** and **5** against the bacterial pathogen, the MBC was also determined. For this compound, the MBC was found to be 16 mg/L against *B. cereus*, *S. typhi*, and P. *aeruginosa*. Compound **4** was more effective against *E. coli* and *B. cereus*, and its MBC value of 8 mg/L was among the smallest. The graph of MBC values for the bacterial organism is shown in Figure 4.

### 2.5. Antifungal Susceptibility

The results of the antifungal activity of uridine derivatives are shown in Table 3 (Appendix A). Interestingly, compounds **5**–**7** significantly inhibited *A. niger* (64 ± 0.44%, 72 ± 0.95%, 66 ± 0.41%) in comparison with the control nystatin (59 ± 0.00). Furthermore, these compounds showed comparable results against *A. flavus.* In summary, these results indicated that uridine derivatives **5**–**7** showed promising antifungal activity against *A. niger* and *A. flavus* [28] (Appendix A).

### 2.6. Anticancer Activity (MTT Assay)

Only compound **6** showed anticancer efficacy after screening compounds **1** to **7**. Finally, the ability of the synthesized compounds to produce cytotoxic effects was investigated by MTT assay. Compound **6** (2′,3′-di-oxo-cinnamoyl-5′-oxo-palmitoyluridine), the most promising compound, was used in vitro on EAC (Ehrlich ascites carcinoma) cells in a dose-dependent manner (Figure 5). At the highest concentration of 500 µg/mL, compound **6** showed only 23.22% cytotoxicity effects, whereas at 250, 125, 62.25, 31.25, and 15.625 µg/mL, it was 14.02%, 7.51%, 5.41%, 4.41%, and 3.11%, respectively. The inhibitory effect of the compound decreased with decreasing concentration, and the IC50 was determined to be 1108.22 g/mL. According to the results, there was no significant reduction in cancer cells.

### 2.7. Structure–Activity Relationship (SAR)

SAR analysis is important for understanding the mechanisms of antimicrobial activity [29] for uridine derivatives (Figure 6). The SAR of uridine analogs was calculated using the antimicrobial activities shown in Table 2 and Table 3. Uridine, **1** exhibited no activity against pathogenic bacteria; consequently, a change in the (uridine, **1**) skeleton had a significant effect on the antibacterial activity. For most of the tested bacteria, fused myristoyl moieties were more active than the uridine molecule. In contrast, trityl-containing analog **5** was weaker than myristoyl derivative **4**. The results of this study indicate that higher concentrations of synthesized compounds were necessary to effectively inhibit the growth of Gram-negative bacteria (with minimum inhibitory concentration values ranging from 0.13 to 4.0 mg/L) compared to Gram-positive bacteria. The distinct behavior was ascribed to the different cell wall structures present in Gram-positive and Gram-negative bacteria. One possible explanation for this phenomenon is that the peptidoglycan layer of Gram-negative bacteria is encased by an outer membrane that limits diffusion due to its lipopolysaccharide (LPS) coating. The LPS layer is a crucial component in facilitating selective permeability [29] and functions as an effective impediment against the swift infiltration of diverse compounds. Hydrophobic interactions may arise among the acyl chains of uridine **1** that are present in the lipid-like nature of bacterial membranes. Due to hydrophobic interactions, bacteria experience a loss of membrane permeability, which ultimately leads to their demise [30]. Similarly, antifungal activities are also observed to increase.

It is postulated that the acyl chains of uridine, which are contained within the lipid-like structure of bacterial membranes, may engage in hydrophobic interactions that are analogous in nature. Bacteria lose membrane permeability as a result of hydrophobic contact, which ultimately results in the death of the bacteria. The study aimed to establish the structure–activity relationship (SAR) of the uridine derivatives that were synthesized based on the results obtained. In addition, it was discovered that adding the palmitoyl group at the C-5′ position and the palmitoyl group at the C-3′ position in compound **2** both improved the antibacterial activity of uridine (**1**) (Figure 7). Additionally, the produced nucleoside analogs showed encouraging binding affinity against both membrane and DNA proteins in the computational analysis. For the convenience of cellular metabolism, nucleoside derivatives mimic natural nucleosides and insert themselves into DNA and RNA. Most antimicrobial nucleoside derivatives function by preventing the chain of nucleic acids from lengthening during the synthesis of DNA (for DNA viruses) or RNA (for RNA viruses).

### 2.8. Cytotoxic Activity of Uridine Derivatives

Figure 8 represents the cytotoxicity of uridine derivatives (**2**–**7**) that were synthesized, as measured by the brine shrimp lethality bioassay method [31]. The figure displays the percentage of shrimp mortality after 24 and 48 h. Long alkyl chains and phenyl rings increased hydrophobicity and cytotoxicity [32]. Based on the data, it was determined that uridine derivative **2** (5′-oxo-palmitoyluridine) had the lowest degree of toxicity, with a 31.21% death. Derivatives **3** (2′,3′-di-oxo-lauroyl-5′-oxo-palmitoyluridine) and **4** (2′,3′-di-oxo-myristoyl-5′-oxo-palmitoyluridine) demonstrated the highest levels of toxicity (i.e., 48.24–49.28% death), showing increased mortality.

In addition, uridine derivatives 5′-oxo-palmitoyl-2′,3′-di-oxo-trityluridine (**5**) and 2′,3′-di-oxo-(4-t-butylbenzoyl)-5′-oxo-palmitoyluridine (**7**) were shown to be less harmful to brine shrimp (34.25% and 33.71% death, respectively). The observation revealed that benzoyl derivatives have a lesser cytotoxic effect than alkyl chain derivatives. In addition, the cytotoxic effect of alkyl chain derivatives increases with increasing concentration.

### 2.9. Predicted Biological Properties (PASS)

The biological properties of uridine derivatives **2**–**7** were predicted using the PASS web server. Table 4 displays the *Pa* and *Pi* values, which are utilized to express the PASS outcomes. Table 2 displays the anticipated outcomes for uridine derivatives **2**–**7**. The antibacterial values ranged from 0.56 < *Pa* < 0.85, antifungal values ranged from 0.22 < *Pa* < 0.75, antiviral values ranged from 0.68 < *Pa* < 0.86, and anticarcinogenic values ranged from 0.43 < *Pa* < 0.82. The results suggest that the aforementioned compounds exhibit greater efficacy against bacteria and viruses in comparison to fungal strains. The antibacterial activity of uridine (**1**, *Pa* = 0.117) was significantly improved (*Pa* = 0.850) by the incorporation of an extra aromatic ring. Conversely, the addition of an aliphatic long chain resulted in a gradual enhancement of its properties. The same observations were seen in the case of antifungal and anticarcinogenic activities, where benzoyl substituents found improved values compared to the chain substituents.

### 2.10. Thermodynamic Study

Free energy and enthalpy assist in determining reaction unpredictability and product stability [33]. Highly negative numbers enhance thermal stability. The dipole moment influences hydrogen bond formation and noncovalent interactions in drug design. Negative free energy (G) encourages natural binding and contact. The uridine derivative had a larger negative value for E, H, and G than the original uridine, suggesting that adding an acyl group may increase these molecules’ interaction and binding with multiple microbial enzymes. Uridine derivative **6** had the highest free energy, enthalpy, and electronic energy (Table 5). High dipole moments indicate polarity [34]. Derivative (**6**) has the highest dipole moment (5.217 Debye) and derivative (**7**) has the lowest (3.254). As the number of substituent carbon atoms increased (**2**–**7**), all metric scores increased. Overall, changing uridine’s hydroxyl (-OH) groups should increase its thermodynamic characteristics [35,36].

### 2.11. Molecular Docking Simulation

To investigate the mode of binding of the synthesized uridine derivatives, three derivatives (**4**, **6**, and **7**) with the maximum binding affinities were chosen based on docking evaluation results. As shown in Table 6, the aromatic ring-based esters showed higher docking energy values than the long chains with derivatives. Collisions between the ligand and active residues of 1RXF and 3000 are described in Figure 9, which was obtained by analyzing the docking complex in Discovery Studio Visualizer. The amino acid residues played a crucial function in the total energy of interaction [37]. Based on docking results, Figure 10 expresses the complex arrangement of the two most active compounds, **6** and **7**.

Thus, substitution of the –OH group by a benzoyl group improved the binding energies compared to the long carbon chain.

The docking results found several types of nonbonding interactions, such as alkyl, pi-alkyl, pi-caton, pi-sigma, pi-donor hydrogen bonds, and pi-pi-stacked interactions with the active site of the target proteins. These results clarify that because of the highly dense electron cloud, the benzoyl ring can smoothly enhance the docking score as well as the inhibitory activity of the uridine derivatives.

Uridine derivatives (**4**, **6**, and **7**) were highly potential ligands (−6.4, −6.9, −6.0, −6.6, −6.8, and −7.8 kcal/mol), and they connected with both proteins via prominent hydrophobic and hydrogen bonding (Table 7 and Table 8). The active sites were specifically located within a hydrogen cleft that is defined by specific amino acid residues, namely, Arg160, Arg162, Arg306, Asn301, Asn304, Tyr184, Thr308, Asp128, Trp297, Gly236, and Gly282. The analyses also found a number of hydrophobic bond contacts with various amino acids, Ile192, Leu204, Val245, Tyr184, Phe283, Phe164, Lys239, Arg242, Arg74, Ala208, Cys281, Trp297, and Lys239. Compounds **6** and **7** featured an additional aromatic ring in the cinnamoyl and 4-*t*-butylbenzoyl groups, resulting in a high electron density in the molecule and the greatest binding affinities.

### 2.12. Molecular Dynamics (MD) Simulation

MD simulation for 400 ns of each system was performed to examine intermolecular conformational binding and interactions along the length of the simulation time. The simulation trajectories were first investigated for root mean square deviation (RMSD) in angstrom [38]. The average RMSD values of the 1RXF-7 and 3OOo-7 complexes were 1.7 Å and 1.5 Å, respectively. Initially, the complexes were unstable in the first 100 ns, followed by stable dynamics toward the end of the simulation time (Figure 11A). Protein residue flexibility was evaluated via root mean square fluctuation (RMSF) [39]. Protein residues were relatively stable with few minor fluctuations (Figure 11B). In particular, the active pocket residues exhibited considerable stability with RMSF values < 2 Å. The RMSD stability of proteins was additionally validated by radius of gyration (RoG) [40], which depicted the highly compact nature of the proteins (Figure 11C). The average RoG value of 1RXF-7 was 36 Å, while that of 3OOo-7 was 35 Å.

### 2.13. Binding Free Energy Calculation

The calculated binding free energy values for the 1RXF-7 complex were −24.7 kcal/mol and −25.72 kcal/mol using the MM-GBSA and MM-PBSA methods, respectively. The overall binding free energy of the 3000-7 complex was determined to be −26.31 kcal/mol (MM-GBSA) and −31.18 kcal/mol (MM-PBSA). The dominant contributor to the general binding energy was the van der Waals energy, with the electrostatic energy following closely behind. The solvation energy made an insignificant contribution to the overall net energy, as reported in [31,33]. Table 9 presents the variances in the binding free energies of compound **7** to the receptors.

### 2.14. Pharmacokinetic Profile and Drug-Likeness Exploration

The pharmacokinetic properties of a drug refer to its absorption, distribution, metabolism, and elimination within the body. The prediction of the compounds was carried out utilizing the pkCSM ADMET descriptors algorithm protocol. The process of drug absorption is influenced by various factors, including but not limited to membrane permeability, which can be assessed by examining the colon cancer cell line (Caco-2), intestinal absorption, skin permeability thresholds, and the presence of a substrate or inhibitor of P-glycoprotein. A level of intestinal absorbance that falls below 30% may indicate inadequate absorption. As shown in Table 10, most of the derivatives show notable absorption scores where the absorption values are >30. A drug will easily enter the skin if the log Kp value is greater than −2.5 cm/h. Table 10 shows that the skin permeability (Kp) of uridine derivatives is −1.00 to −2.33 cm/h (<−2.5). Thus, all derivatives exhibited excellent skin penetrability. In addition, as shown in Table 10, the Caco-2 permeability (log Papp) results for uridine derivatives ranged from 0.268 to 0.854 cm/s, log Papp < 0.9 cm/s, indicating that these compounds have a low Caco-2 permeability [41].

High water solubility is favorable for providing adequate quantities of active components in such small volumes of pharmaceutical dosage. Table 10 shows that the tested (uridine, **1**) analogs were soluble.

It was found that the low VDss was <0.71 l/kg (log VDss < −0.15) and the high was >2.81 (log VDss > 0.45). VDss values for (uridine, **1**) derivatives ranged from −0.598 to 0.182. Table 11 reveals that the logPS value of uridine derivatives were between –3.65 and −2.23, logPS < −3 (excluding **2** and **3**); hence, derivatives (**4**–**7**) cannot penetrate the CNS. The pkCSM pharmacokinetics model estimated the total clearance log (CLtot) of a synthesized chemical in log (mL/min/kg). Table 11 demonstrates that uridine derivative log CLtot values ranged from 0.279 to 1.434 mL/min/kg, which might predict their excretion [42].

## 3. Materials and Methods

### 3.1. General Information

Electrothermal melting point apparatus temperatures were uncorrected. Thin-layer chromatography (TLC) on Kieselgel GF_254_ detected spots by spraying with 1% H_2_SO_4_ and heating at 150–200 °C. Column chromatography used silica gel G_60_. Using a Fourier transform infrared (FTIR) spectrophotometer (IR Prestige-21, Shimadzu, Kyoto, Japan), infrared spectrum analyses were recorded at the Department of Chemistry, University of Chittagong. ^1^H-NMR (400 MHz) and ^13^C-NMR (100 MHz) spectra were recorded at the Wazed Miah Science Research Centre, JU, and Dhaka, Bangladesh. Liquid chromatography–electrospray ionization tandem mass spectrometry in positive ionization mode produced mass spectra of manufactured substances. A Büchi rotary evaporator was used for all evaporations. Figure 12 shows a schematic flow of the work plan.

### 3.2. Synthesis of Uridine Derivatives

#### 5′-Oxo-palmitoyluridine (**2**)

In a round bottle flask, uridine (1) (200 mg, 0.82 mmol) in anhydrous pyridine (3 mL) was chilled to 0 °C, and palmitoyl chloride (0.20 mL, 1.1 molar eq.) was added. After 6–7 h at 0 °C, the reaction mixture was left overnight at room temperature with constant stirring. TLC (methanol-chloroform, 1:16) showed complete conversion of the starting material into a faster-moving single product (R*_f_* = 0.50). Work-up as usual and purification by silica gel column chromatography with methanol-chloroform (1:16 as the elutant) yielded palmitoyl derivative (2) (179.4 mg, 45%) as a crystalline solid. Palmitoyl derivatives (2) crystallized as needles from chloroform-*n*-hexane, m.p. 64–65 °C.

Yield = 45%. FTIR (KBr) (ν_max_): 1702 cm**^−^**^1^ (C=O), 3408–3511 cm**^−^**^1^ (br, -OH). δ_H_ (ppm) 9.0 (1H, s, -NH), 7.28 (1H, m, H-6), 6.85 ((1H, d, J = 5.4 Hz, H-1′), 6.78 (1H, s, 2′-OH), 6.01 (1H, dd, J = 2.2 and 12.0 Hz, H-5′a), 5.46 (1H, dd, J = 2.3 and 12.0 Hz, H-5′b), 5.05 (1H, s, 3′-OH), 4.77 (1H, d, J = 8.2 Hz, H-5), 4.46 (1H, d, J = 5.6 Hz, H-2′), 4.38 (1H, dd, J = 7.4 and 5.5 Hz, H-3′), 4.31 (1H, m, H-4′), 2.37 {2H, m, CH_3_(CH_2_)_12_CH_2_C*H*_2_CO-}, 1.64 {2H, m, CH_3_(CH_2_)_12_C*H*_2_CH_2_CO-}, 1.28 {24H, m, CH_3_(C*H*_2_)_12_CH_2_CH_2_CO-}, 0.90 {6H, m, C*H*_3_(CH_2_)_14_CO-}. ^13^C NMR (100 MHz, CDCl_3_): *δ*_C_ 172.21 {CH_3_(CH_2_)_14_*C*O-}, 164.09 (C-4), 150.80 (C-2), 140.43 (C-6), 101.29 (C-5), 88.25 (C-1′), 86.74 (C-4′), 82.71 (C-3′), 72.44 (C-2′), 60.47 (C-5′), 34.43, 34.38, 31.95, 31.91(×2), 29.52, 29.31, 29.11, 25.11 (×2), 24.77, 22.61, 21.54, 20.01 {CH_3_(*C*H_2_)_14_CO-}, 14.11 {*C*H_3_(CH_2_)_14_CO-}. LC–MS [M+1]^+^ 483.62. Anal. Calcd. for C_25_H_42_O_7_N_2_: C = 62.22, H = 8.77; found: C = 62.24, H = 8.78%.

### 3.3. General Procedure for the Synthesis of 2′,3′-Di-oxo-acyl Uridine Derivatives (**3**–**7**)

Lauroyl chloride (0.19 mL, 4 molar eq.) was added to a cooled (0 °C) stirred solution of palmitoyl derivative (**2**) (99.2 mg, 0.21 mmol) in anhydrous pyridine (3 mL); stirred at 0 °C for 8 h and then stored overnight at room temperature. TLC (methanol-chloroform, 1:16) revealed full conversion of the reactant into one product (R*_f_* = 0.51). The reaction mixture was processed as usual after adding a few pieces of ice to the reaction flask to eliminate excess reagent. Percolation of the syrup through a silica gel column with methanol-chloroform (1:20, as elutant) yielded the lauroyl derivative (**3**) (80 mg, 56.98%) as a solid product, which could be recrystallized (chloroform-*n*-hexane) to yield the title product (**3**), needles, m.p. (58–60 °C).

Compound **4** (86 mg) as needles, m.p. 44–46 °C (myristoyl derivatives), **5** (350 mg) as needles, m.p. 142–145 °C (trityl derivatives), **6** (162.2 mg) as needles, m.p. 59–61 °C (cinnamoyl derivatives), and **7** (53 mg) as needles, m.p. 54–56 °C (4-*t*-butylbenzoyl derivatives) were prepared using a similar reaction and purification method.

#### 3.3.1. 2′,3′-Di-*oxo*-lauroyl-5′-oxo-palmitoyluridine (**3**)

Yield = 59.98%. FTIR (KBr) (ν_max_): 1701 cm^−1^ (C=O). δ_H_ (ppm) 9.0 (1H, s, -NH), 7.19 (1H, d, J = 7.2 Hz, H-6), 6.77 (1H, d, J = 5.2 Hz, H-1′), 6.41 (1H, dd, J = 2.0, and 12.0 Hz, H-5′a), 5.89 (1H, dd, J = 2.0, and 12.0 Hz, H-5′b), 5.71 (1H, d, J = 8.1 Hz, H-5), 5.57 (1H, d, J = 5.4 Hz, H-2′), 5.22 (1H, m, H-3′), 4.44 (1H, m, H-4′), 2.38 {2H, m, CH_3_(CH_2_)_12_CH_2_C*H*_2_CO-}, 2.35 {4H, m, 2 × CH_3_(CH_2_)_9_C*H*_2_CO-}, 1.67 {2H, m, CH_3_(CH_2_)_12_C*H*_2_CH_2_CO-}, 1.65 {4H, m, 2 × CH_3_(CH_2_)_8_C*H*_2_CH_2_CO-}, 1.30 {24H, m, CH_3_(C*H*_2_)_12_CH_2_CH_2_CO-}, 1.29 {32H, m, 2 × CH_3_(C*H*_2_)_8_CH_2_CH_2_CO-}, 0.89 {6H, m, 2 × C*H*_3_(CH_2_)_10_CO-}, 0.90 {6H, m, C*H*_3_(CH_2_)_14_CO-}. ^13^C NMR (100 MHz, CDCl_3_): *δ*_C_ 172.50, 172.46 {2 × CH_3_(CH_2_)_10_*C*O*-*}, 172.43 {CH_3_(CH_2_)_14_*C*O-}, 164.11 (C-4), 150.23 (C-2), 140.48 (C-6), 101.31 (C-5), 88.64 (C-1′), 86.72 (C-4′), 82.01 (C-3′), 72.12 (C-2′), 60.54 (C-5′), 34.38, 31.90 (×2), 29.52 (×2), 29.32 (×3), 29.10, 25.08 (×2), 24.32, 22.15 (×3), 22.05, 21.72, 21.21, 20.11 (×2) {2 × CH_3_(*C*H_2_)_10_CO-}, 13.53, 13.49 {2 × *C*H_3_(CH_2_)_10_CO-}, 34.41, 34.23, 31.22, 31.92, 31.23, 29.50. LC–MS [M + 1]^+^ 848.23. Anal. Calcd. for C_49_H_86_O_9_N_2_: C = 69.47, H = 10.23; C = 69.49, H = 10.25%.

#### 3.3.2. 2′,3′-Di-*oxo*-myristoyl-5′-oxo-palmitoyluridine (**4**)

Yield = 53.43%. FTIR (KBr) (ν_max_): 1706 cm^−1^ (C=O). δ_H_ (ppm) 9.2 (1H, s, -NH), 7.23 (1H, m, H-6), 6. 82 (1H, d, J = 5.1 Hz, H-1′), 6.61 (1H, dd, J = 2.0, and 12.0 Hz, H-5′a), 6.17 (1H, dd, J = 2.0, and 12.0 Hz, H-5′b), 5.78 (1H, d, J = 8.1 Hz, H-5), 5.66 (1H, d, J = 5.1 Hz, H-2′), 5.48 (1H, dd, J = 7.1, and 5.1 Hz H-3′), 4.54 (1H, m, H-4′), 2.37 {2H, m, CH_3_(CH_2_)_12_CH_2_C*H*_2_CO-}, 2.32 {4H, m, 2 × CH_3_(CH_2_)_11_C*H*_2_CO-}, 1.64 {2H, m, CH_3_(CH_2_)_12_C*H*_2_CH_2_CO-}, 1.63 {4H, m, 2 × CH_3_(CH_2_)_10_C*H*_2_CH_2_CO-}, 1.28 {24H, m, CH_3_(C*H*_2_)_12_CH_2_CH_2_CO-}, 1.27 {40H, m, 2 × CH_3_(C*H*_2_)_10_CH_2_CH_2_CO-}, 0.90 {6H, m, 2 × C*H*_3_(CH_2_)_12_CO-}, 0.89 {6H, m, C*H*_3_(CH_2_)_14_CO-}.4. ^13^C NMR (100 MHz, CDCl_3_): *δ*_C_ 172.54, 172.40 {2 × CH_3_(CH_2_)_12_*C*O-}, 172.20 {CH_3_(CH_2_)_14_*C*O-}, 164.22 (C-4), 150.85 (C-2), 141.11 (C-6), 101.44 (C-5), 88.20 (C-1′), 86.33 (C-4′), 82.43 (C-3′), 72.41 (C-2′), 60.42 (C-5′), 34.41, 34.33, 31.90, 31.34(×2), 29.34, 29.21, 29.10, 25.09 (×2), 24.21, 22.55, 21.11, 20.21 {CH_3_(*C*H_2_)_14_CO-}, 34.38, 34.36, 34.12 (×2), 31.92, 31.90 (×2), 29.59 (×2), 29.32 (×3), 29.15, 25.01 (×3), 24.96, 22.67 (×3), 21.72, 21.69, 20.09 (×2) {2 × CH_3_(*C*H_2_)_12_CO-}, 14.09 {*C*H_3_(CH_2_)_14_CO-}, 14.08, 14.01 {2 × *C*H_3_(CH_2_)_12_CO-}. LC–MS [M + 1]^+^ 904.34. Anal. Calcd. for C_53_H_94_O_9_N_2_: C = 70.47, H = 10.49; found: C = 70.48, H = 10.48%.

#### 3.3.3. 5′-*oxo*-Palmitoyl-2′,3′-di-oxo-trityluridine (**5**)

Yield = 49.89%. FTIR (KBr) (ν_max_): 1703 cm^−1^ (C=O). δ_H_ (ppm) 9.0 (1H, s, -NH), 7.61 (1H, d, J = 7.3 Hz, H-6), 7.60 (2 × 6H, m, Ar-H), 7.44 (2 × 9H, m, Ar-H), 6.18 (1H, d, J = 5.2 Hz, H-1′), 6.0 (1H, dd, J = 2.0, and 12.0 Hz, H-5′a), 5.85 (1H, dd, J = 2.0, and 12.0 Hz, H-5′b), 5.51 (1H, d, J = 8.1 Hz, H-5), 5.41 (1H, d, J = 5.4 Hz, H-2′), 5.38 (1H, m, H-3′), 4.51 (1H, m, H-4′), 2.39 {2H, m, CH_3_(CH_2_)_12_CH_2_C*H*_2_CO-}, 1.68 {2H, m, CH_3_(CH_2_)_12_C*H*_2_CH_2_CO-}, 1.31 {24H, m, CH_3_(C*H*_2_)_12_CH_2_CH_2_CO-}, 0.92 {6H, m, C*H*_3_(CH_2_)_14_CO-}. ^13^C NMR (100 MHz, CDCl_3_): *δ*_C_ 172.09 {CH_3_(CH_2_)_14_*C*O-}, 164.18 (C-4), 150.11 (C-2), 140.65 (C-6), 101.70 (C-5), 88.33 (C-1′), 86.61 (C-4′), 82.31 (C-3′), 72.40 (C-2′), 60.11 (C-5′), 145.47 (×4), 145.24 (×3), 145.05 (×4), 129.61 (×6), 129.33 (×4), 127.84 (×4), 127.21 (×3), 127.09 (×3), 126.57 (×5) {(2 × *C*_6_H_5_)_3_C-}, 81.62, 81.31{2×(C_6_H_5_)_3_*C*-}, 34.40, 34.11, 31.82, 31.11(×2), 29.01, 29.63, 29.51, 25.11(×2), 24.12, 22.45, 21.31, 20.26 {CH_3_(*C*H_2_)_14_CO-}, 14.14 {*C*H_3_(CH_2_)_14_CO-}, LC-MS [M + 1]^+^ 968.26. Anal. Calcd. for C_63_H_70_O_7_N_2_: C = 78.23, H = 7.29; found: C = 78.24, H = 7.26%.

#### 3.3.4. 2′,3′-Di-*oxo*-cinnamoyl-5′-oxo-palmitoyluridine (**6**)

Yield = 64.96%. FTIR (KBr) (ν_max_): 1701 cm^−1^. δ_H_ (ppm) 8.98 (1H, s, -NH), 8.10 (4H, m, Ar-H), 7.81 (1H, d, J = 7.1 Hz, H-6), 7.57, 7.54 (2 × 1H, 2 × d, J = 16.0 Hz, 2 × PhC*H*=CHCO-), 7.28 (6H, m, Ar-H), 7.22 (1H, d, J = 5.3 Hz, H-1′), 7.10 (1H, dd, J = 2.0, and 12.0 Hz, H-5′a), 6.28 (1H, dd, J = 2.0, and 12.0 Hz, H-5′b), 6.0, 5.97 (2 × 1H, 2 × d, J = 16.0 Hz, 2 × PhCH=C*H*CO-), 5.71 (1H, d, J = 8.1 Hz, H-5), 5.36 (1H, d, J = 5.2 Hz, H-2′), 5.34 (1H, dd, J = 7.1, and 5.1 Hz H-3′), 4.41 (1H, m, H-4′), 2.37 {2H, m, CH_3_(CH_2_)_12_CH_2_C*H*_2_CO-}, 1.64 {2H, m, CH_3_(CH_2_)_12_C*H*_2_CH_2_CO-}, 1.28 {24H, m, CH_3_(C*H*_2_)_12_CH_2_CH_2_CO-}, 0.90 {6H, m, C*H*_3_(CH_2_)_14_CO-}. ^13^C NMR (100 MHz, CDCl_3_): *δ*_C_ 172.13 {CH_3_(CH_2_)_14_*C*O-}, 165.84, 165.78 (2 × C_6_H_5_CH=CHCO-), 164.65 (C-4), 150.23 (C-2), 140.76 (C-6), 101.11 (C-5), 88.54 (C-1′), 86.41 (C-4′), 82.62 (C-3′), 72.40 (C-2′), 60.23 (C-5′), 150.57, 150.35 (2 × C_6_H_5_CH=CHCO-), 132.99, 132.81 (×2), 132.76, 132.21, 129.20 (×3), 129.12, 129.06 (×3) (2 × C_6_H_5_CH=CHCO-), 122.06, 121.88 (2 × C_6_H_5_CH=CHCO-), 34.41, 34.22, 31.16, 31.01(×2), 29.32, 29.26, 29.13, 25.10 LC–MS [M + 1]^+^ 743.91. Anal. Calcd. for C_43_H_54_O_9_N_2_: C = 69.52, H = 7.32; found: C = 69.54, H = 7.31%.

#### 3.3.5. 2′,3′-Di-*oxo*-(4-t-butylbenzoyl)-5′-oxo-palmitoyluridine (**7**)

Yield = 59.69%. FTIR (KBr) (ν_max_): 1718 cm^−1^ (C=O). δ_H_ (ppm) 9.01 (1H, s, -NH), 7.85 (4H, m, 2 × Ar-2H), 7.51 (4H, m, 2 × Ar-2H), 7.21 (1H, m, H-6), 6.80 (1H, d, J = 5.0 Hz, H-1′), 6.73 (1H, dd, J = 2.2, and 12.0 Hz, H-5′a), 6.17 (1H, dd, J = 2.3, and 12.0 Hz, H-5′b), 5.82 (1H, d, J = 8.1 Hz, H-5), 5.73 (1H, d, J = 5.2 Hz, H-2′), 5.54 (1H, dd, J = 7.0, and 5.1 Hz H-3′), 4.70 (1H, m, H-4′), 2.36 {2H, m, CH_3_(CH_2_)_12_CH_2_C*H*_2_CO-}, 1.65 {2H, m, CH_3_(CH_2_)_12_C*H*_2_CH_2_CO-}, 1.38, 1.36 {18H, 2 × s, 2 × (C*H*_3_)_3_C-}, 1.26 {24H, m, CH_3_(C*H*_2_)_12_CH_2_CH_2_CO-}, 0.91 {6H, m, C*H*_3_(CH_2_)_14_CO-}. ^13^C NMR (100 MHz, CDCl_3_): *δ*_C_ 174.40, 174.23 {2 × (CH_3_)_3_CC_6_H_4_*C*O-}, 172.15 {CH_3_(CH_2_)_14_*C*O-}, 164.32 (C-4), 150.67 (C-2), 140.33 (C-6), 101.23 (C-5), 88.61 (C-1′), 86.55 (C-4′), 82.66 (C-3′), 72.39 (C-2′), 60.16 (C-5′), 132.44, 132.40(×2), 132.40, 130.94 (×2), 129.91 (×3), 126.52, 125.50 (×2) {2 × (CH_3_)_3_C*C*_6_H_4_CO-}, 35.60, 35.57 {(2×)(CH_3_)_3_*C*C_6_H_4_CO-}, 13.67 (×2), 13.65 (×2), 13.42 (×2) {(2×)(*C*H_3_)_3_CC_6_H_4_CO-}, 34.40, 34.24, 31.47, 31.44(×2), 29.16, 29.08, 29.07, 25.23(×2), 24.55, 22.62, 21.52, 20.11 {CH_3_(*C*H_2_)_14_CO-}, 14.09 {*C*H_3_(CH_2_)_14_CO-}. LC–MS [M + 1]^+^ 804.05. Anal. Calcd. for C_47_H_66_O_9_N_2_: C = 70.29, H = 8.28; found: C = 70.31, H = 8.26%.

### 3.4. In Vitro Antimicrobial Experiment

For the antimicrobial assessment, five human pathogenic bacteria, and two plant pathogenic fungi were used. The results of this analysis are presented in Appendix A.

#### 3.4.1. Antibacterial Susceptibility

The antibacterial activity of the synthesized compound was evaluated using the disc diffusion method in accordance with the clinical and laboratory standards institute’s [43] recommendations. Molten Mueller–Hinton agar and Sabouraud dextrose agar media were inoculated with the inoculum at 35 ± 2 °C and then poured into a Petri dish (90 mm). Each sample was soaked on 6 mm absorbent discs in dimethyl sulfoxide (DMSO) and placed on the inoculation plates. After 2 h at 4 °C, the plates were incubated at 35 ± 2 °C for 18–20 h. The inhibition zone diameters were determined. Positive control discs of selected antibiotics were included in each assay; inhibition zones were compared to the reference discs. Negative controls were treated with DMSO, and the assays were performed in triplicate.

#### 3.4.2. Determination of Minimum Inhibitory Concentration (MIC) and Minimum Bactericidal Concentration (MBC)

Microdilution techniques were employed to calculate the minimum inhibitory concentration (MIC) and the minimum bactericidal concentration (MBC). A serial dilution of the tested compound in pure DMSO was performed at concentrations between 0.125 and 128 μg/mL and incorporated into semisolid agar medium. After solidification, the medium incorporated was seeded from suspensions of bacteria at a concentration of 0.5 McFarland standard (1.0 × 10^8^ CFU/mL). After 18–20 h of incubation at 35 ± 2 °C, the absence or presence of microbial growth in Petri dishes was determined by examination with the naked eye. MIC was interpreted as the smallest concentration of uridine esters inhibiting any visible growth, and MBC was defined as the minimum bactericidal concentration that can kill the organism. MIC and MBC were determined only with microorganisms that displayed important inhibitory zones. All compounds were evaluated three times against each microorganism. The solvent control DMSO did not show any antimicrobial activity.

#### 3.4.3. Screening of Mycelial Growth

The potato dextrose agar (PDA)-based antifungal experiment was screened using the food poisoning approach [44]. Sterilized 70 mm glass Petri dishes were filled with 20 mL melted PDA (45 °C). After the medium solidified, sterilized needles placed small amounts of mycelia from the two fungal pathogens in the center of each plate. The synthesized uridine (**1**) analogs were tested on the mycelial growth of the two fungi using the ‘poisoned food’ method. After 48 h at 37 °C, mycelial growth was measured.

### 3.5. Experimental Animals and Ethical Clearance

Ethical and animal Swiss albino mice from ICDDR’B were obtained. The Institutional Animal, Medical Ethics, Biosafety and Biosecurity Committee (IAMEBBC) for Experimentations on Animals, Humans, Microbes, and Living Natural Sources (102(6)/320-IAMEBBC/IBSc), Institute of Biological Sciences (IBSc), University of Rajshahi, Bangladesh, approved the in vivo experiment.

### 3.6. Antiproliferative Activity against EAC Cells by MTT Assay

Ehrlich ascites carcinoma (EAC) cells were cultivated in conventional DMEM from a Swiss albino mouse bearing a tumor [45]. A 96-well flat-bottomed cell culture plate with 2 × 10^4^ cells in 100 μL was incubated at 37 °C in a CO_2_ incubator for 24 h. Controls were three wells of untreated EAC cells in DMEM. EAC cells were treated for 48 h with chemical 6 doses of 37.5–600 μg/mL. After removing the aliquot, 10 mM PBS and 20 μL of MTT reagent (5 mg/mL MTT in PBS) were added. After another 8 h at 37 °C, the aliquot was drained, and acidic isopropanol (200 μL) was added. Absorbance values at 570 nm were recorded after another 30 min CO_2_ incubator incubation. The following formula was used to estimate cell proliferation incubation:Proliferation inhibition ratio (%) = (A − B) × 100/A
where A and B represent the cellular homogenate OD570 nm from control wells and compound 6-treated wells, respectively.

### 3.7. Growth Inhibition Assay of Ehrlich ascites Carcinoma (EAC) Cells

To produce ascites tumors, 18 mice were injected intraperitoneally with 0.1 mL of 99% viable tumor cells. These cells survived the 0.4% trypan blue exclusion experiment. Using a hemocytometer, 200 μL of saline containing EAC cells was injected into 18 mice. Each group had six mice: “Control”, “low-dose”, and “high-dose”. Compound **6** was injected at 2.0 and 4.0 mg/kg/day into “low-dose” and “high-dose” animals 24 h after cell inoculation. Mice from both groups were slaughtered 24 h after the last dose after five days of therapy. Each mouse’s EAC cells were collected in 0.9% saline and counted by a hemocytometer. The following formula was used to calculate the chemical EAC cell growth inhibition percentages:Percentage of inhibition = 100 − {(cells from compound 574 − treated mice/cells from control mice) × 100}.

### 3.8. Structure–Activity Relationship (SAR) Study

Structure–activity relationship (SAR) analysis identified the resulting molecule’s active component. The Hunt [46] and Kim [47] membrane permeation concept is applied in drug design employing this popular approach.

### 3.9. Cytotoxic Activity Evaluation

Uridine derivative toxicity was tested using the brine shrimp lethality assay (BSLA) [48]. Each vial contained 5 mL of NaCl solution and 20 μL of uridine derivatives dissolved in DMSO. Vials A, B, C, and D contained 4, 8, 16, and 32 μL of material. Each vial received 10 brine shrimp nauplii at three concentrations. Ten nauplii in 5 mL saltwater were used for a control test. The vials were incubated at room temperature for 24–48 h. After incubation, the vials were magnified and counted to determine how many survived. Each concentration has an average nauplii mortality rate. No controls died.

### 3.10. PASS Enumeration and Bioactivity

PASS (http://www.pharmaexpert.ru/passonline/ (accessed on 30 January 2023) was used to determine the antibacterial activity spectrum of selected (uridine, **1**) derivatives [49]. Initially, the (uridine, **1**) analog structures were sketched and then transformed into their smile forms using the famous SwissADME free online application (http://www.swissadme.ch (accessed on 30 January 2023) to determine the antimicrobial spectrum with the PASS web tool.

### 3.11. Geometry DFT Optimization

Quantum mechanical computations determine thermal, molecular orbital, and molecular electrostatic parameters in computational drug design [50]. Gaussian 09 optimized geometry and modified all synthesized analogs [51]. Density functional theory (DFT) with Beck’s (B) three-parameter hybrid model, Lee, Yang, and Parr’s (LYP) correlation functional, and the 3–21 G basis set optimized and estimated their thermal and molecular orbital parameters [52,53].

### 3.12. Protein Selection and Molecular Docking

The protein data bank presented *E. coli* (1RXF) and *S. typhi* (3000) pdb structures [54]. PyMol (version 1.3) stripped all heteroatoms and water molecules [55]. Swiss-PdbViewer (4.1.0) diminished protein energy [56]. Finally, molecular docking simulation [57,58] was executed employing PyRx (version 0.8) to visualize the target protein as a macromolecule and the (uridine, **1**) analogs as ligands. Accelrys Discovery Studio (version 4.1) was used to analyze docking results and anticipate nonbonding interactions between (uridine, **1**) analogs and receptor protein amino acid chains [59].

### 3.13. MD Simulations

The structural minimization and dynamics simulation of the selected 1RXF-7 complex and 3000-7 complex were accomplished using AMBER20 software (2022). The force fields ff14SB [60] and GAFF [61] were employed to generate parameters for enzymes and compounds, respectively. Tleap was thought to add hydrogen atoms to complexes. After water solvation, counter ions neutralized the complexes. Periodic boundary conditions simulated complexes. The particle-mesh Ewald procedure [62] handled long-distance electrostatic interactions with an 8 Å cutoff for nonbonding interactions. Water molecules were energy reduced 1500 times, and the total system was 1000 times. The system temperature was gently increased to 300 K, and equilibration took 150 ps at normal temperature and pressure. MD simulation manufacturing ran at 300 K and 1 atm pressure for 400 ns. Langevin dynamics [63] controlled temperature, and the SHAKE algorithm [64] constrained hydrogen-atom covalent bonds. AMBER’s CPPTRAJ module [65] plotted complicated stability.

### 3.14. Binding Free Energy Calculation

The binding free energy of the complexes was estimated using the MM/GBSA and MM/PBSA methods [66]. The net energy value was based on 1000 frames from simulation trajectories.

### 3.15. Pharmacokinetic and Drug-Likeness Prediction

Drug development ADMET property prediction is vital for preventing clinical failure. Thus, the best-identified esters were evaluated using pkCSM [67] for their in silico pharmacokinetics parameters to prevent clinical trial failure and increase their chances of being selected as potential candidate drugs. Its absorption in the intestine, percolation in the blood–brain barrier and central nervous system, and metabolism suggest chemical biotransformation, drug clearance, and toxicity.

## 4. Conclusions and Future Perspectives

In this study, uridine analogs were developed, synthesized, and tested in vitro and in silico for antibacterial, cytotoxic, physicochemical, molecular docking, MD simulations, and pharmacokinetic properties. Most uridine analogs were high-yielding and antibacterial. Adding aliphatic chains and aromatic rings to the uridine **1** structure increased its biological activity. 4-*t*-Butylbenzoyl and cinnamoyl derivatives (**6** and **7**) had improved pharmacokinetic and biological spectra and stronger bacterial and fungal activity. Molecular docking showed that uridine derivatives (**2**–**7**) had promising antibacterial effectiveness against *E. coli* (1RXF) and *S. typhi* (3000). In silico, derivatives **4**, **6**, and **7** inhibited target microorganisms. MD simulations at 400 ns supported the docked complex’s binding stability in trajectory analysis. Most developed compounds have enhanced kinetic characteristics and preserved drug-likeness rules. Therefore, the research presented here suggests uridine analogs as antimicrobial/anticancer agents.

## Data Availability

Data are available in this article.

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
