# Peer review of "Uridine Derivatives: Synthesis, Biological Evaluation, and In Silico Studies as Antimicrobial and Anticancer Agents"

_medicina, 2023, doi:10.3390/medicina59061107_

Round 1

Reviewer 1 Report

In the manuscript ”medicina-2389379-peer-review-v1” with the title Fused Pyrimidine Derivatives: Synthesis, Biological Evaluation, and In Silico Studies as Antimicrobial and Anticancer Agents the authors developed, synthesized, and tested in vitro and in silico six uridine analogs.

 The objectives were clearly stated and explained in the manuscript. The manuscript is overall well written and has good organization with minor English language and style spell check required.

Mostly, I found the manuscript to be compelling and of high quality in its presentation. Still, I pose some comments hoping to be useful for the authors:

The resolution and quality of some Figures are low, the authors should provide higher quality figures, especially for Figure 10.

The compound 7 from Figure 10B should have different coloured atoms, instead of the green ones.

In addition, in the caption of Figure 10, I recommend explaining the colour code used to represent the ligand-receptor bonding.

What were the criteria for choosing 1RXF and 3O00 to be used in the molecular docking procedure?

How were they prepared for the docking procedure?

How was the molecular docking procedure validated?

The manuscript is overall well written and has good organization with minor English language and style spell check required.

Author Response

List of corrections of reviewer comments

medicina-2389379

N.B.: All the corrections have been made in the text.

Reviewer-1

  1. The resolution and quality of some Figures are low, the authors should provide higher quality figures, especially for Figure 10.

Response: We have increased the resolution and quality of all figures.

  1. The compound 7 from Figure 10B should have different coloured atoms, instead of the green ones. In addition, in the caption of Figure 10, I recommend explaining the colour code used to represent the ligand-receptor bonding.

Response: Unfortunately, our laptop is not supported now. Thank you, we will follow your nice suggestion in our forthcoming manuscripts.

  1. What were the criteria for choosing 1RXF and 3O00 to be used in the molecular docking procedure?

Response: Dear Sir, we have used both bacterial 1RXF and 3O00 for molecular docking. Please have a look at the following:

  1. According to in vitro test results, including antibacterial activity, compounds showed promising antifungal activity. So, we tried to maintain alignment and comparison with both target
  2. According to the structure-activity relationship (SAR) study, we have assumed that electron-withdrawing groups are responsible for antifungal activity.
  3. Finally, In molecular docking, we found significant binding energy and interactions between the compounds and fungal targets.

Overall, to rationalize all the observations we docked against both bacterial and fungal targets.

  1. How were they prepared for the docking procedure?

Response: We have optimized the compounds by Gaussian 09 software and saved as pdb file. Then the pdb file interacted with the protein in PyRx software. After that, the ligand and protein were combined in PyMol software. Finally, the results were analysed by Discovery Studio software.

  1. How was the molecular docking procedure validated?

Response: For docking validation, we have extracted the structure ligands of both proteins 1RXF and 3O00. Then we re-docked both ligands with the proteins. The re-docked ligand structure was superimposed to its initial structure to check the RMSD value. Finally, following these criteria, we have performed the docking of all compounds.

  1. The manuscript is overall well written and has good organization with minor English language and style spell check required.

Response: Corrections have been done.

Reviewer 2 Report

The following corrections should be made to a manuscript submitted for review:

1. Correct the structures in figure 1 (no double bond in pyrimidine).

2. In scheme 1, correct the structure (there should be no methyl group in position 5).

3. Why does Scheme 1 say "Palatino Linotype"?

4. Correct the names of unions. There should not be -O- in the names, only oxo.

5. Correct the numbering of groups in compounds (we number the rings from heteroatoms, in oxolan from oxygen).

6. There should be no H5'a numbering, H5'b (this is not a group in oxolan) this is a CH2 group outside the ring.

7. Correct the structure of derivative 2 in figure 8.

8. What is the solvent in 1 HNMR?

9. Why in anal calcd. there is no nitrogen?

Author Response

List of corrections of reviewer comments

medicina-2389379

N.B.: All the corrections have been made in the text.

Reviewer-2

  1. Correct the structures in figure 1 (no double bond in pyrimidine).

Response: Thank you very much for your comment. Corrections have been done.

  1. In scheme 1, correct the structure (there should be no methyl group in position 5).

Response: Corrections have been done. We deleted (–methyl) group.

  1. Why does Scheme 1 say "Palatino Linotype"?

Response: We removed it

  1. Correct the names of unions. There should not be -O- in the names, only oxo.

Response: Corrections have been done.

  1. Correct the numbering of groups in compounds (we number the rings from heteroatoms, in oxolan from oxygen).

Response: We have denoted the atoms of six member ring (nitrogenous base) as 1, 2, 3, 4, 5, 6 etc. and the five-member ring (sugar) as 1´, 2´, 3´, 4´ 5´ etc.

  1. There should be no H5'a numbering, H5'b (this is not a group in oxolan) this is a CH2 group outside the ring.

Response: We have denoted the atoms of six member ring (nitrogenous base) as 1, 2, 3, 4, 5, 6 etc. and the five-member ring (sugar) as 1´, 2´, 3´, 4´ 5´ etc. As there are two protons in 5´ position, we denoted them as H5'a and H5'b.

  1. Correct the structure of derivative 2 in figure 8.

Response: Corrections have been done.

  1. What is the solvent in 1 HNMR?

Response: The solvent is Chloroform (CHCl3)

  1. Why in anal calcd. there is no nitrogen?

Response: The “N” atom is available in the parent compound (nitrogenous base) which is fixed and we didn’t introduce any “N” atom in the parent structure. So, there is no “N” to be calculated.

Reviewer 3 Report

#Title:

 “Fused Pyrimidine Derivatives” should be replaced with “Uridine derivatives” or “Modified uridine”.

#Abstract: This section has to be structurally revised. The common structure of background, methods, results, and conclusion should be followed.  

-          Line 20: the IC50 value of 1108.22 g/mL seems irrational.

#Keywords: more effective keywords can be selected.

# Introduction: needs to be revised. This section covers general issues and cannot highlight the importance and necessity of the investigation.

-          Lines 54-55: “Uridine and related … brain and spinal cord”. This statement is not related to the context sentences.

-          The section contains excessive/non-specific citations.

# Results:

-          Figure 2 and Scheme 1 should be moved to the material and methods section.

 # References: The number of articles cited in this manuscript is very high. Besides, the authors must support their work with a wide range of related studies worldwide. An overwhelming self-citation was detected in the study.

Author Response

List of corrections of reviewer comments

medicina-2389379

N.B.: All the corrections have been made in the text.

Reviewer-3

  1. #Title: “Fused Pyrimidine Derivatives” should be replaced with “Uridine derivatives” or “Modified uridine”.

Response: Thank you so much! Corrections have been done.

  1. #Abstract: This section has to be structurally revised. The common structure of background, methods, results, and conclusion should be followed.  

Response: Corrections have been done.

  1. Line 20: the IC50 value of 1108.22 g/mL seems irrational.

Response: We have removed it.

  1. #Keywords: more effective keywords can be selected.

Response: Corrections have been done.

  1. # Introduction: needs to be revised. This section covers general issues and cannot highlight the importance and necessity of the investigation.

Response: Corrections have been done.

  1. Lines 54-55: “Uridine and related … brain and spinal cord”. This statement is not related to the context sentences.

Response: We have removed it.

  1. The section contains excessive/non-specific citations.

Response: We have added citations.

  1. # Results:   Figure 2 and Scheme 1 should be moved to the material and methods section.

Response: We transferred the Figure 2 to the material and methods section. But the Scheme 1 should be kept at the results section as it represents synthetic results.

  1. # References: The number of articles cited in this manuscript is very high. Besides, the authors must support their work with a wide range of related studies worldwide. An overwhelming self-citation was detected in the study.

Response: Corrections have been done. We deleted the followings:

            Self-citation has been deleted as below:

  1. Kawsar, S.M.A.; Hosen, M.A.; Alam, A.; Islam, M.; Ferdous, J.; Fujii, Y.; Ozeki, Y. Thymidine Derivatives as Inhibitors against Novel Coronavirus (SARS-CoV-2) Main Protease: Theoretical and Computational Investigations. Adv. Chem. Res. 2021, 69, 89–129.
  2. Kawsar, S.M.A.; Hamida, A.A.; Sheikh, A.U.; Hossain, M.K.; Shagir, A.C.; Sanaullah, A. F.M.; Manchur, M.A.; Hasan, I.; Ogawa, Y.; Fujii, Y.; Koide, Y.; Ozeki, Y. Chemically Modified Uridine Molecules Incorporating Acyl Residues to Enhance Antibacterial and Cytotoxic Activities. Int. J. Org. Chem. 2015, 5, 232–245.
  3. Kawsar, S.M.A.; Islam, M.; Jesmin, S.; Manchur, M.A.; Hasan, I.; Rajia, S. Evaluation of the Antimicrobial Activity and Cytotoxic Effect of Some Uridine Dderivatives. Int. J. Biosci. 2018, 12, 211–219.
  4. Kawsar, S.M.A.; Hosen, M.A.; Chowdhury, T.S.; Rana, K.M.; Fujii, Y.; Ozeki, Y. Thermochemical, PASS, Molecular Docking, Drug-Likeness and In Silico ADMET Prediction of Cytidine Derivatives Against HIV-1 Reverse Transcriptase. Rev. de Chimie. 2021, 72, 159–178.
  5. Kawsar, S.M.A.; Hosen, M.A.; El Bakri, Y.; Ahmad, S.; Sopi, T.A.; Goumri-Said, S. In Silico Approach for Potential Antimicrobial Agents Through Antiviral, Molecular Docking, Molecular Dynamics, Pharmacokinetic and Bioactivity Predictions of Galactopyranoside Derivatives. Arab J. Basic. Appl. Sci. 2022, 29, 99–112.
  6. Kawsar, S.M.A.; Hosen, M.A.; Ahmad, S.; El Bakri, Y.; Laaroussi, H.; Hadda, T.B.; Almalki, F.A.; Ozeki, Y.; Goumri-Said, S. Potential SARS-CoV-2 RdRp Inhibitors of Cytidine Derivatives: Molecular Docking, Molecular Dynamic Simulations, ADMET, and POM Analyses for the Identification of Pharmacophore Sites. PLoS ONE, 2022, 17, e0273256.
  7. Kawsar, S.M.A.; Kabir, A.K.M.S.; Bhuiyan, M.M.R.; Siddiqa, A.; Anwar, M.N. Synthesis, Spectral and Antimicrobial Screening Studies of Some Acylated D-Glucose Derivatives. Rajiv Gandhi Univ. Health Sci. J. Pharm. Sci. 2012, 2, 107–115.
  8. Kawsar, S.M.A.; Islam, M.M.; Chowdhury, S.A.; Hasan, T.; Hossain, M.K.; Manchur, M.A.; Ozeki, Y. Design and Newly Synthesis of Some 1,2-O-Isopropylidene-α-D-Glucofuranose Derivatives: Characterization and Antibacterial Screening Studies. Hacettepe J. Biol. Chem. 2013, 41, 195–206.
  9. Kawsar, S.M.A.; Almalki, F.A.; Hadda, T.B.; Laaroussi, H.; Khan, M.A.R.; Hosen, M.A.; Mahmud, S.; Aounti, A.; Maideen, N.M.P.; Heidarizadeh, F.; Soliman S.S.M. Potential Antifungal Activity of Novel Carbohydrate Derivatives Validated by POM, Molecular Docking and Molecular Dynamic Ssimulations Analyses. Mol. Simul. 2023, 49, 60–75.
  10. Kawsar, S.M.A.; Ouassaf, M.; Hosen, M.A.; Chtita, S.; Qais, F.A.; Belaidi, S. Physicochemical, ADMET, Molecular Docking and Molecular Dynamics Simulations against Bacillus subtilis HmoB for Antibacterial Potentiality of Methyl α-D-Glucopyranoside Derivatives. Philippine J. Sci. 2022, 151, 1393–1417.

11    Kawsar, S.M.A.; Ouassaf, M.; Chtita, S.; Jui, A.B.; Belaidi, S. PASS Prediction, Molecular Docking and Pharmacokinetic Studies of Acyl Substituted Bioactive Galactopyranoside Esters as Antibacterial Agents. Macedonian J. Chem. Chem. Eng. 2022, 41, 47–64.

Other 3 references also deleted as below:

  1. Izaguirre, J.A.; Catarello, D.P.; Wozniak, J.M.; Skeel, R.D. Langevin Stabilization of Molecular dynamics. J. Chem. Phys. 2001, 114, 2090–2098.
  2. Roe, D.R.; Cheatham III, T.E. PTRAJ and CPPTRAJ: Software for Processing and Analysis of Molecular Dynamics Trajectory Data. J. Chem. Theory Comput. 2013, 9, 3084–3095.
  3. Genheden, S.; Ryde, U. The MM/PBSA and MM/GBSA Methods to Estimate Ligand-Binding Affinities. Expert Opin. Drug Discov. 2015, 10, 449–461.

Round 2

Reviewer 3 Report

The authors have addressed the comments. However, there is still a high percentage of unnecessary self-citation. I am sure there are many relevant articles published by other researchers worldwide that can be cited to support the present study, instead of authors' previous works.

Please take into account that supporting your findings/arguments just with your previous works can weaken the novelty and credibility of your work. 

Can be improved. 

Author Response

List of corrections of reviewer comments

medicina-2389379

2nd Time Reviewer’s Comments:

[N.B.: All the corrections have been made in the text with highlighted].

Reviewer-3

Comment: The authors have addressed the comments. However, there is still a high percentage of unnecessary self-citation. I am sure there are many relevant articles published by other researchers worldwide that can be cited to support the present study, instead of authors' previous works.

Please take into account that supporting your findings/arguments just with your previous works can weaken the novelty and credibility of your work. 

Response: Thank you very much for the comment. We reduced the citations and again deleted our Self-citations references. But few citations must be kept in the Text because which are very much related to our study.

We deleted our own citation References as below:

  1. Rana, K.M.; Ferdous, J.; Hosen, A.; Kawsar, S.M.A. Ribose Moieties Acylation and Characterization of Some Cytidine Analogs. J. Siberian Fed. Univ. Chem. 2020, 13, 465–478.
  2. Kawsar, S.M.A.; Hamida, A.A.; Sheikh, A.U.; Hossain, M.K.; Shagir, A.C.; Sanaullah, A. F.M.; Manchur, M.A.; Hasan, I.; Ogawa, Y.; Fujii, Y.; Koide, Y.; Ozeki, Y. Chemically Modified Uridine Molecules Incorporating Acyl Residues to Enhance Antibacterial and Cytotoxic Activities. Int. J. Org. Chem. 2015, 5, 232–245.
  3. Rana, K.M.; Maowa, J.; Alam, A.; Dey, S.; Hosen, A.; Hasan, I.; Fujii, Y.; Ozeki, Y.; Kawsar, S.M.A. In Silico DFT Study, Molecular Docking, and ADMET Predictions of Cytidine Analogs with Antimicrobial and Anticancer Pproperties. In Silico Pharmacol. 2021, 9, 1–24.
  4. Kawsar, S.M.A.; Islam, M.; Jesmin, S.; Manchur, M.A.; Hasan, I.; Rajia, S. Evaluation of the Antimicrobial Activity and Cytotoxic Effect of Some Uridine Dderivatives. Int. J. Biosci. 2018, 12, 211–219.
  5. Hosen, M.A.; Munia, N.S.; Al-Ghorbani, M.; Baashen, M.; Almalki, F.A.; Hadda, T.B.; Ali, F.; Mahmud, S.; Saleh, M.A.; Laaroussi, H.; Kawsar, S.M.A. Synthesis, Antimicrobial, Molecular Docking and Molecular Dynamics Studies of Lauroyl Thymidine Analogs against SARS-CoV-2: POM Study and Identification of the Pharmacophore Sites. Bioorg. Chem. 2022, 125, 105850.
  6. Munia, N.S.; Hosen, M.A.; Khaldun, M.A.; Al-Ghorbani, M.; Baashen, M.;  Hossain, M.K.; Ali, F.; Mahmud, S.; Shimu, M.S.S.; Almalki, F.A.;  Hadda, T.B.; Laaroussi, H.; Naimi, S.; Kawsar, S.M.A. Synthesis, Antimicrobial, SAR, PASS, Molecular Docking, Molecular Dynamics and Pharmacokinetics Studies of 5´-O-Uridine Derivatives Bearing Acyl Moieties: POM Study and Identification of the Pharmacophore Sites. Nucleos. Nucleot. Nucleic Acids. 2022, 41, 1036–1083.
  7. Alam, A.; Rana, K.M.; Hosen, M.A.; Dey, S.; Bezbaruah, B.; Kawsar, S.M.A. Modified Thymidine Derivatives as Potential Inhibitors of SARS-CoV: PASS, In Vitro Antimicrobial, Physicochemical and Molecular Docking Studies. Phys. Chem. Res. 2022, 10, 391–409.
  8. Maowa, J.; Alam, A.; Rana, K.M.; Dey, S.; Hosen, A.; Fujii, Y.; Hasan, I.; Ozeki, Y.; Kawsar, S.M.A. Synthesis, Characterization, Synergistic Antimicrobial Properties and Molecular Docking of Sugar Modified Uridine Derivatives. Ovidius Univ. Ann. Chem2021, 32, 6–21.
  9. Arifuzzaman, M.; Islam, M.M.; Rahman, M.M.; Mohammad, A.R.; Kawsar, S.M.A. An Efficient Approach to the Synthesis of Thymidine Derivatives Containing Various Acyl Groups: Characterization and Antibacterial Activities. ACTA Pharm. Sci. 2018, 56, 7–22.
  10. Sanjida, J.; Sumi, R.D.; Rahman, M.; Mariam, I.; Kanaly, A.R.; Fuji, Y.; Hayashi, N.; Ozeki, Y.; Kawsar, S.M.A. An Efficient Synthesis and Spectroscopic Characterization of Some Uridine Derivatives. J. Bang. Chem. Soc. 2017, 29, 12–20.
  11. Bulbul, M.Z.H.; Hosen, M.A.; Ferdous, J.; Misbah, M.M.H.; Kawsar, S.M.A. Thermochemical, DFT Study, Physicochemical, Molecular Docking and ADMET Predictions of Some Modified Uridine Derivatives. Int. J. New Chem. 2021, 8, 88–110.
  12. Kawsar, S.M.A.; Hosen, M.A.; Alam, A.; Islam, M.; Ferdous, J.; Fujii, Y.; Ozeki, Y. Thymidine Derivatives as Inhibitors against Novel Coronavirus (SARS-CoV-2) Main Protease: Theoretical and Computational Investigations. Adv. Chem. Res. 2021, 69, 89–129.
  13. Tasneem, S.C.; Jannatul, F.; Misbah, M.M.H.; Bulbul, M.Z.H.; Kawsar, S.M.A. Partial Acylation of Pyrimidine Thymidine Derivatives. J. Bang. Chem. Soc. 2019, 31, 40–48.
  14. Hosen, M.A.; Alam, A.; Islam, M.; Fujii, Y.; Ozeki. Y.; Kawsar, S.M.A. Geometrical Optimization, PASS Prediction, Molecular Docking, and In Silico ADMET Studies of Thymidine Derivatives against FimH Adhesin of Escherichia coli. Bulg. Chem. Commun. 2021, 53, 327–342.
  15. Shamsuddin, T.; Hosen, M.A.; Alam, M.S.; Emran, T.B.; Kawsar, S.M.A. Uridine Derivatives: Antifungal, PASS Outcomes, ADME/T, Drug-Likeliness, Molecular Docking and Binding Energy Calculations. Med. Sci. Int. Med. J. 2021, 10, 1373–1386.
  16. Kawsar, S.M.A.; Hosen, M.A.; Chowdhury, T.S.; Rana, K.M.; Fujii, Y.; Ozeki, Y. Thermochemical, PASS, Molecular Docking, Drug-Likeness and In Silico ADMET Prediction of Cytidine Derivatives Against HIV-1 Reverse Transcriptase. Rev. de Chimie. 2021, 72, 159–178.
  17. Kawsar, S.M.A.; Hosen, M.A.; El Bakri, Y.; Ahmad, S.; Sopi, T.A.; Goumri-Said, S. In Silico Approach for Potential Antimicrobial Agents Through Antiviral, Molecular Docking, Molecular Dynamics, Pharmacokinetic and Bioactivity Predictions of Galactopyranoside Derivatives. Arab J. Basic. Appl. Sci. 2022, 29, 99–112.
  18. Alam, A.; Hosen, M.A.; Islam M.; Ferdous, J.; Fujii, Y.; Ozeki. Y.; Kawsar, S.M.A. Synthesis, Antibacterial and Cytotoxicity Assessment of Modified Uridine Molecules. Curr. Adv. Chem. Biochem. 2021, 6, 114–129.
  19. Kawsar, S.M.A.; Hosen, M.A.; Ahmad, S.; El Bakri, Y.; Laaroussi, H.; Hadda, T.B.; Almalki, F.A.; Ozeki, Y.; Goumri-Said, S. Potential SARS-CoV-2 RdRp Inhibitors of Cytidine Derivatives: Molecular Docking, Molecular Dynamic Simulations, ADMET, and POM Analyses for the Identification of Pharmacophore Sites. PLoS ONE, 2022, 17, e0273256.
  20. Ahmmed, F.; Islam, A.U.; Mukhrish, Y.E.; Bakri, Y.El.; Ahmad, S.; Ozeki, Y.; Kawsar, S.M.A. Efficient Antibacterial/Antifungal Activities: Synthesis, Molecular Docking, Molecular Dynamics, Pharmacokinetic and Binding Free Energy of Galactopyranoside Derivatives. Molecules 2023, 28, 219.
  21. Chowdhury, T.S.; Ferdous, J.; Bulbul, M.Z.H.; Misbah, M.M.H.; Dey, S.; Hasan, I.; Kawsar, S.M.A. Antimicrobial and Anticancer Activities of some Partial Acylated Thymidine Derivatives. J. Bio. Sci. 2021, 29, 11–22.
  22. Islam, S.; Hosen, M.A.; Ahmad, S.; ul Qamar, M.T.; Dey, S.; Hasan, I.; Fujii, Y.; Ozeki, Y.; Kawsar, S.M.A. Synthesis, Antimicrobial, Anticancer Activities, PASS Prediction, Molecular Docking, Molecular Dynamics and Pharmacokinetic Studies of Designed Methyl α-D-Glucopyranoside Esters. J. Mol. Struct. 2022, 1260, 132761.
  23. Amin, M.R.; Yasmin, F.; Hosen, M.A.; Dey, S.; Mahmud, S.; Saleh, M.A.; Hasan, I.; Fujii, Y.; Yamada, M.; Ozeki, Y.; Kawsar, S.M.A. Synthesis, Antimicrobial, Anticancer, PASS, Molecular Docking, Molecular Dynamic Simulations and Pharmacokinetic Predictions of Some Methyl β-D-Galactopyranoside Analogs. Molecules 2021, 26, 1–25.
  24. Amin, M.R.; Yasmin, F.; Dey, S.; Mahmud, S.; Saleh, M.A.; Emran, T.B.; Hasan, I.; Rajia, S.; Ogawa, Y.; Fujii, Y.; Yamada, M.; Ozeki, Y.; Kawsar, S.M.A. Methyl β-D-Galactopyranoside Esters as Potential Inhibitors for SARS-CoV-2 Protease Enzyme: Synthesis, Antimicrobial, PASS, Molecular Docking, Molecular Dynamics Simulations and Quantum Computations. Glycoconjugate J. 2021, 38, 1–30.
  25. Kawsar, S.M.A.; Hasan, T.; Chowdhury, S.A.; Islam, M.M.; Hossain, M.K.; Mansur, M.A. Synthesis, Spectroscopic Characterization and In Vitro Antibacterial Screening of Some D-Glucose Derivatives. Int. J. Pure App Chem. 2013, 8, 125–135.
  26. Kawsar, S.M.A.; Mamun, S.M.A.; Rahman, M.S.; Yasumitsu, H.; Ozeki, Y. In Vitro Antibacterial and Antifungal Effects of a 30 kDa D-Galactoside-Specific Lectin from the Demosponge, Halichondria okadai. Int. J. Biol. Life Sci. 2011, 6, 31–37.
  27. Hosen, M.A.; Bakri, Y.E.; Rehman, H.M.; Hashem, H.E.; Saki, M.; Kawsar, S.M.A. A Computational Investigation of Galactopyranoside Esters as Antimicrobial Agents Through Antiviral, Molecular Docking, Molecular Dynamics, Pharmacokinetics, and Bioactivity Prediction. J. Biomol. Struct. Dyn. 2023, 41, 1–16.
  28. Kawsar, S.M.A.; Kabir, A.K.M.S.; Bhuiyan, M.M.R.; Siddiqa, A.; Anwar, M.N. Synthesis, Spectral and Antimicrobial Screening Studies of Some Acylated D-Glucose Derivatives. Rajiv Gandhi Univ. Health Sci. J. Pharm. Sci. 2012, 2, 107–115.
  29. Hosen, M.I.; Mukhrish, Y.E.; Jawhari, A.H.; Celik, I.; Erol, M.; Abdallah, E.M.; Al-Ghorbani, M.; Baashen, M.; Almalki, F.A.; Laaroussi, H.; Hadda, T.B.; Kawsar, S.M.A. Design, Synthesis, In Silico and POM Studies for the Identification of the Pharmacophore Sites of Benzylidene Derivatives. Molecules. 2023, 28, 2613.
  30. Kawsar, S.M.A.; Almalki, F.A.; Hadda, T.B.; Laaroussi, H.; Khan, M.A.R.; Hosen, M.A.; Mahmud, S.; Aounti, A.; Maideen, N.M.P.; Heidarizadeh, F.; Soliman S.S.M. Potential Antifungal Activity of Novel Carbohydrate Derivatives Validated by POM, Molecular Docking and Molecular Dynamic Ssimulations Analyses. Mol. Simul. 2023, 49, 60–75.
  31. Kawsar, S.M.A.; Ouassaf, M.; Hosen, M.A.; Chtita, S.; Qais, F.A.; Belaidi, S. Physicochemical, ADMET, Molecular Docking and Molecular Dynamics Simulations against Bacillus subtilis HmoB for Antibacterial Potentiality of Methyl α-D-Glucopyranoside Derivatives. Philippine J. Sci. 2022, 151, 1393–1417.
  32. Kawsar, S.M.A.; Ouassaf, M.; Chtita, S.; Jui, A.B.; Belaidi, S. PASS Prediction, Molecular Docking and Pharmacokinetic Studies of Acyl Substituted Bioactive Galactopyranoside Esters as Antibacterial Agents. Macedonian J. Chem. Chem. Eng. 2022, 41, 47–64.

Newly added References in the Text:

  1. Thomson, J.M.; Lamont, I.L. Nucleoside Analogues as Antibacterial Agents. Front. Microbiol. 10, 2019, 952.
  2. Zenchenko, A.A.; Drenichev, M.S.; Il’icheva, I.A. Antiviral and Antimicrobial Nucleoside Derivatives: Structural Features and Mechanisms of Action. Mol. Biol. 55, 2021, 786–812.
  3. Koszalka, G.W.; Daluge, S.M.; Boyd, F.L. Advances in Nucleoside and Nucleotide Antiviral Therapies. Ann. Rep. Med. Chem. 1998, 33, 163–171.
  4. Chan, K.Y.; Kinghorm, A.B.; Hollenstein, M.; Tanner, J.A. Chemical Modifications for a Next Generation of Nucleic Acid Aptamers. Chem. Bio. Chem. 2022. 23, e202200006.
  5. Mikhailopulo, I.A.; Miroshnikov, A.I. New Trends in Nucleoside Biotechnology. Acta Nat. 2010, 2, 36–58.
  6. Arbour, C.A.; Imperiali, B. Uridine Natural Products: Challenging TargetS AND Inspiration for Novel Small Molecule Inhibitors. Bioorg. Med. Chem. 2020, 28, 115661.
  7. Thatipamula, R.K.; Narsimha, S.; Battula, K.; Chary, V.R.; Mamidala, E.; Reddy, N.V. Synthesis, Anticancer and Antibacterial Evaluation of Novel (Isopropylidene) Uridine[1,2,3]triazole Hybrids. J. Saudi Chem. Soc. 2017, 21, 795–802.
  8. Brab, A.W.; Leavy, T.M.; Robins, L.I.; Guan, Z.; Six, D.A.; Zhou, P.; Bertozzi, C.R.; Raetz, C.R.H. Uridine-based Inhibitors as New Leads for Antibiotics Targeting E. coli LpxC. Biochemistry 2009, 48, 3068-3077.
  9. Saha S, Banerjee S, Ganguly S. Molecular docking studies of some novel hydroxamic acid derivatives. Int J Chem Tech Res. 2010;2:932–36.
  10. Kim, E.; Kang, W. Pharmacokinetics of Uridine Following Ocular, Oral and Intravenous Administration in Rabbits. Biomol. Ther. 2013, 21, 170–172.
